# Relationship between Lower Extremity Peripheral Arterial Disease and Mild Cognitive Impairment in Hemodialysis Patients

**DOI:** 10.3390/jcm12062145

**Published:** 2023-03-09

**Authors:** Akinori Nishimura, Sumi Hidaka, Takayuki Kawaguchi, Aki Watanabe, Yasuhiro Mochida, Kunihiro Ishioka, Milanga Mwanatanbwe, Takayasu Ohtake, Shuzo Kobayashi

**Affiliations:** 1Rehabilitation Unit, Shonan Kamakura General Hospital, Okamoto 1370_1, Kamakura 247-8533, Japan; 2Kidney Disease and Transplant Center, Shonan Kamakura General Hospital, Kamakura 247-8533, Japan; 3Shonan Research Institute of Innovative Medicine (sRIIM), Kamakura 247-8533, Japan; 4Department of Community Mental Health and Law, National Institute of Mental Health, National Center of Neurology and Psychiatry, Kodaira 187-8553, Japan; 5School of Allied Health Sciences, Kitasato University, Sagamihara 252-0373, Japan; 6Department of Pathology, University of Mbuji-Mayi, Mbuji-Mayi 8010, Democratic Republic of the Congo; 7International Division of Tokushukai of Medical Corporation, Tokushukai, Chiyoda-ku 102-0074, Japan

**Keywords:** hemodialysis, mild cognitive impairment, peripheral artery disease, lower extremity artery disease, ankle brachial index, toe brachial index

## Abstract

Background: The link between arterial stiffness and mild cognitive impairment (MCI) in patients on hemodialysis (HD) has been receiving increased attention. The purpose of this study was to investigate the relationship between cognitive function and ankle brachial index (ABI) and toe brachial index (TBI) values in patients on hemodialysis. Of the 100 participants (mean age: 67.9 years; average history of hemodialysis: 7.3 years). Of these, 46.0% had MCI. The MoCA-J scores were significantly higher in the ABI ≥ 1.06 group. However, the MoCA-J scores divided into the two groups according to the TBI cutoff value were not significantly different. In a multiple regression model with the MoCA-J scores as the objective variable, the ABI was a significantly associated factor. This study indicates that a low ABI might be associated with MCI.

## 1. Introduction

The aging of the hemodialysis (HD) population in our country is evident, and dementia is becoming a major problem. Early detection of mild cognitive impairment (MCI), which is considered a transitional stage preceding clinical dementia, is critical. The prevalence of MCI in patients with end-stage renal failure [1,2] is 51–76%. Furthermore, we previously reported in a multicenter study that 58.2% of patients on HD had MCI [3]. The definition of dementia, according to ICD-10, is an acquired decline in cognitive domains that interferes with daily life and social interaction [4]. Cognitive impairment in patients on HD leads to decreased self-care skills and difficulty in physical activity, as well as activities of daily living (ADL) [5]. Screening assessments such as the Mini-Mental State Examination (MMSE) may underestimate cognitive impairment; therefore, it is important to detect MCI early and efficiently before it reaches the stage of overt dementia [1,6,7].

The causes of cognitive dysfunction in patients on dialysis are complex, including not only aging, hypertension, diabetes, and smoking, but also anemia, inflammation, hyperhomocysteinemia, malnutrition, and dialysis-related hypotension [8,9]. The most common complication in patients on dialysis is cerebro-cardiovascular disease due to atherosclerosis, ischemic heart disease, cerebral infarction, and peripheral artery disease (PAD) [10,11,12]. PAD is commonly used as a designation for upper and lower extremity arterial disease. However, in this study, we referred to it as lower extremity artery disease (LEAD), because we included only arterial disease of the lower extremities. LEAD is often asymptomatic [13] and is an independent risk factor for death during induction dialysis [14]. Previously, we reported a 24.3% incidence of LEAD in patients undergoing HD [15]. The survival rates among patients on dialysis with ischemic foot ulcers are 65.2% and 23.4% at 1 and 5 years, respectively, with a poor prognosis [16]. LEAD affects not only the lower extremities but also the cardiovascular system [17,18]. The patients with severe LEAD tend to have poorer cognitive function than the controls [19,20]. Extremity blood pressure measurement is the most convenient and useful method of functional assessment and noninvasive diagnosis of LEAD in clinical practice [21]. The ankle brachial index (ABI) and toe brachial index (TBI) are frequently used to screen for LEAD. Generally, an ABI value of 0.90 or less is considered to be associated with LEAD; however, the cutoff value in patients on HD is 1.06 or less [22]. Moreover, the measurement of the TBI is considered important to assess peripheral circulatory impairment.

A previous study reported that a low ABI was associated with low cognitive function in patients on HD [23]. However, the ABI evaluates relatively large arteries at the ankle to diagnose LEAD, while the TBI evaluates peripheral arteries of the lower extremities. It is well known that LEAD is more likely to occur in relatively distal arteries in dialysis patients [24]. The purpose of this study was to clarify the relationship between LEAD and cognitive function in patients on HD using ABI and TBI values.

## 2. Methods

### 2.1. Study Population

This study assessed 101 outpatients on maintenance HD who attended our hospital between February and March 2017. The exclusion criteria were neurological diseases such as overt dementia and Parkinson’s disease, a history of psychiatric disorders, such as depression and schizophrenia, decreased health status due to malignancy or serious infections; and visual, hearing, or speech impairment.

### 2.2. Protocol

This was a single-center, retrospective, cross-sectional study. MCI was assessed using the Japanese version of the Montreal Cognitive Assessment (“MoCA-J”) as the MCI screening test and a hand grip strength measurement before the start of dialysis [3]. The ABI and TBI values within 6 months of the measurement date, blood data from the most recent pre-dialysis period, and the patient’s background data were collected from electronic medical records.

### 2.3. MoCA-J

The MoCA-J, which is a relatively new assessment scale developed by Nasreddine et al. [25]. in 2005 and translated by Fujiwara et al. [26] in 2010, was used as a screening test to assess MCI. The 30-item scale consists of cube-copying, clock-drawing, naming, five-word learning, delayed playback, number-counting, selective attention, calculation, sentence repetition, word recall, and disorientation. MoCA scores of 21 or lower showed a high predictive ability for severe cognitive impairment, with a sensitivity of 86%, specificity of 55%, and negative predictive accuracy of 91% [27]. These tasks cover the cognitive domains of visuospatial/executive, naming tasks, attention, language, abstract conceptual thinking, memory, and orientation. In the Japanese community, the MoCA-J showed a sensitivity of 93.0% and a specificity of 87.0% when the cutoff was less than 26 points [26]. The MoCA-J was assessed before dialysis by a trained occupational therapist.

### 2.4. Hand Grip Strength

Hand grip strength was measured using a digital grip strength meter (T.K.K. 5401 GRIP D; Takei Scientific Instruments Co., Ltd., Niigata, Japan), while in a standing position with the right and left upper limbs hanging down from the body, twice in each position, and the measurement with the higher value was adopted as the hand grip strength.

### 2.5. Measurement of ABI and TBI

The ABI and TBI were measured 3090 min before the start of dialysis using a VaSera VS-1000 (Fukuda Denshi Co., Ltd., Bunkyo, Japan). The blood pressure was measured on the non-arteriovenous fistula side in the arm, both ankles, and both toes using an oscillometric method. The ratio of ankle systolic pressure divided by upper arm systolic pressure was calculated as the ABI, and the lower value of the ankle systolic pressure was used. The ratio of toe systolic blood pressure divided by upper arm systolic blood pressure was calculated as the TBI, and the lower value of the toe systolic blood pressure was used.

### 2.6. Analysis

Normality in each survey item was confirmed using the Shapiro–Wilk test. In the comparison of the two groups, a Student’s t-test was performed for items for which normality was confirmed, and Wilcoxon’s rank sum test was rejected. The χ^2^ test or Fisher’s exact test was used to examine the differences in gender, primary disease, complications, and ABI values.

Next, we compared the MoCA-J subitems of the two groups with ABI values of less than and more than 1.06, and the MoCA-J subitems with TBI values of less than and more than 0.6. Normality in each survey item was confirmed using the Shapiro–Wilk test. In the comparison of each group, the Student’s t-test was used for items when normality was confirmed, and Wilcoxon’s rank sum test was used for non-normal distribution.

In order to determine the factors involved in the cognitive function of patients’ ABI and TBI values, we conducted a single regression model of ABI or TBI values against MoCA-J scores. Multiple regression models by stepwise methods were conducted with MoCA-J scores as the objective variable and ABI or TBI values as the explanatory variables. The explanatory variables included the ABI/TBI, sex, age, body mass index, history of maintenance hemodialysis, hand grip strength, pulse pressure, pulse rate, and serum albumin. In addition, multiple regression analysis was performed using the ABI and TBI values as explanatory variables, and standardized regression coefficients were calculated to compare the magnitude of the effects of the ABI and TBI values on the MoCA-J scores. Finally, subgroup analyses were conducted to evaluate the association between the ABI or TBI and MoCA-J scores for each subgroup, classified by gender or the presence of diabetes mellitus, cerebrovascular disease (denoted as stroke, including cerebral hemorrhage and cerebral infarction), or myocardial infarction. A *p*-value of less than 0.05 was considered statistically significant. The statistical analyses were conducted using R (version 4.2.2, R Foundation for Statistical Computing, Vienna, Austria).

## 3. Results

Table 1 provides an overview of the study participants. The analysis included 100 participants (67 male (67.0%), age 67.9 ± 11.2 years, and history of dialysis 7.3 ± 6.8 years); one participant was excluded due to unknown TBI laboratory values. The primary diseases included diabetic nephropathy in 44 patients (44.0%) and non-diabetic nephropathy in 56 patients (56.0%). Of the patients with non-diabetic nephropathy as the primary disease, six participants (10.7%) developed diabetes. The MoCA-J score was 24.6 ± 4.4 (9–30). A total of 46 patients (46.0%) had MoCA-J scores of less than 26 points and were diagnosed with MCI. The clinical characteristics of the two groups are summarized in Table 1. The MCI group was older and had higher pulse pressure, a slower pulse rate, lower grip strength, and lower albumin than the No-MCI group. Furthermore, the MCI group showed lower ABI and TBI values than the No-MCI group.

### 3.1. Comparison of MoCA-J Divided by Cutoffs for ABI and TBI Values

Figure 1 illustrates the total MoCA-J scores for the two groups, ABI < 1.06 and ABI ≥ 1.06 and TBI < 0.6 and TBI ≥ 0.6. The MoCA-J scores for TBI < 0.6 and TBI ≥ 0.6 were 23.6 ± 5.0 and 25.1 ± 3.9, respectively, and were not significantly different between the two groups (*p* = 0.170).

In the MoCA-J subtests, the two groups with ABI < 1.06 and ABI ≥ 1.06 showed significant reductions in visuospatial and executive systems, attention, language, abstract concepts, and delayed replay (*p* < 0.05) (Table 2). However, there were no predominant differences in the subitems between the two groups with TBI < 0.6 and TBI ≥ 0.6.

### 3.2. Relationship between MoCA-J Scores and ABI and TBI

The results of the single and multivariate regression analyses are summarized in Table 3. In the single regression model of MoCA-J scores and ABI in hemodialysis patients, the ABI was an associated factor for MoCA-J (β = 6.940, 95% CI: 3.083 to 10.797, *p* = 0.001) and for the TBI (β = 7.127, 95% CI: 2.378 to 11. 876, *p* = 0.004).

A stepwise multiple regression analysis with the MoCA-J scores as the objective variable and the ABI and TBI values as explanatory variables was performed as a multivariate linear regression model. With the ABI as the explanatory variable, the MoCA-J score indicated a significant relationship between the ABI (β = 4.742 95% CI: 1.411 to 8.073, *p* = 0.006), age (β = −0.125, 95% CI: −0.207 to −0.044, *p* =0.003), and hand grip strength (β = 0.1904, 95% CI: 0.028 to 0.3514, *p* = 0.022). When the TBI was used as an explanatory variable, the TBI (β = 4.656 95% CI: 0.56 to 8.743, *p* = 0.026), age (β = −0.125, 95% CI: −0.208 to −0.041, *p* = 0.004), BMI (β = −0.147, 95% CI: −0.341 to −0.046, *p* = 0.040), and hand grip strength (β = 0.220, 95% CI: 0.057 to 0.382, *p* = 0.009) showed significant associations with the MoCA-J score. 

In a multiple regression model with the ABI and TBI values as explanatory variables for the MoCA-J scores, the ABI (β = 5.310, 95% CI: 0.566 to 10.054, standard β = 0.260, *p* = 0.029) was a significantly associated factor, while the TBI (β = 3.376, 95% CI: −2.362 to 9.113, stdβ = 0.136, *p* = 0.246) was not a significantly associated factor. Furthermore, the standardized partial regression coefficient for the ABI was greater than that for the TBI.

### 3.3. Subgroup Analysis of the Association between ABI/TBI and MoCA-J Score

The subgroup analysis with gender and baseline comorbidities showed that the ABI was positively associated with the MoCA-J except in men, patients with diabetes, patients with complications of stroke, and patients with no history of coronary arterial disease (Figure 2). Similarly, the TBI was shown to be positively associated with the MoCA-J, except in men, patients with diabetes, patients with complications of stroke (including cerebral hemorrhage and cerebral infarction), and those with coronary arterial disease (Figure 2).

## 4. Discussion

The purpose of this study was to evaluate the relationship between LEAD and MCI in patients on HD. MCI was assessed by the MoCA, and LEAD was assessed by the ABI and TBI. The results revealed that 46.0% of participants on maintenance HD were diagnosed with MCI, and the ABI values and TBI values were significantly related to the MoCA-J scores. Furthermore, lower ABI values were associated with lower MoCA-J scores. We believe it is useful to investigate the possibility of cognitive decline in patients diagnosed with LEAD based on their ABI values. Our results also support those of previous studies that have shown an association between a lower ABI and MCI as assessed by MoCA-J [23].

Compared to the Japanese patients on HD in a study from 2018, the participants of this study had a mean age of 67.9 years, a history of dialysis of 7.3 years, and diabetes as the primary disease in 39.0% of cases [28], which represents the current status of dialysis patients in Japan. The prevalence of MCI in this study was 46.0%, which was comparable to the 48.9% reported in a previous prospective cohort study of dialysis patients without overt dementia [29]. The ABI value of less than 1.06 had an incidence of 31.0%, which is lower than that reported for Japanese patients with diabetes on maintenance dialysis, at 37.2% [30]. The smaller proportion of diabetic patients among the studied patients on HD may have influenced the results.

The causes of cognitive dysfunction in patients on dialysis are complex and include traditional risk factors (including age, hypertension, diabetes, and smoking), non-traditional risk factors (including anemia, albuminuria, inflammation, homocysteinemia, and malnutrition), and dialysis-related factors (hypotension) [8,9]. In addition, the brain and kidneys are low-resistance end organs exposed to large volumes of blood flow and share common anatomic and vascular regulating characteristics that make them susceptible to vascular injury [31]. The cause of cognitive dysfunction in patients with chronic renal failure is influenced by cerebrovascular disease, which is associated with many cardiovascular risk factors [32]. We reported that hippocampal atrophy correlated significantly with hyperhomocysteinemia in patients on HD, and regional cerebral blood flow was reduced in all of the patients on HD, irrespective of clinical symptoms or MMSE scores [33,34]. Low ABI values, which assess ischemia in the lower extremities, also correlate with cortical thinning and reduced cortical thickness in the limbic, parietal, temporal, and occipital lobes [35]. Furthermore, patients with severe LEAD tend to have poorer cognitive function than the controls [19,20], supporting the association of a low ABI with cognitive decline in this study.

In this study, the comparison of MoCA-J scores divided by cutoffs for the ABI and TBI values showed that an ABI ≥ 1.06 was associated with significantly higher MoCA-J scores than those with an ABI < 1.06. However, there was no significant difference in the MoCA-J scores between the TBI < 0.6 and TBI ≥ 0.6. The present study suggests that patients with low ABI levels have significant declines in cognitive function, particularly in visuospatial and executive systems, attention, language, abstract concepts, and delayed reproduction. A stepwise multiple regression analysis including ABI values against MoCA-J scores showed that the ABI values and hand grip strength were positively associated with the MoCA-J scores and aging was negatively associated with them. The multiple regression analysis using the stepwise method, including the TBI values, also showed a positive association between the TBI values and the MoCA-J scores. Furthermore, aging, low BMI, and grip strength were shown to influence MCI. Among patients on HD, lower hand grip strength is significantly associated with a higher prevalence of MCI, particularly among those under the age of 70 with a history of stroke [3]. Multiple regression models using the ABI and TBI as explanatory variables for the MoCA-J scores showed a predominant association with the ABI. These results suggest that while both the ABI and TBI are linked to MCI, the association between the ABI and MCI is stronger than that between the TBI and MCI.

Using a cutoff value of ABI < 0.9, the sensitivity of the ABI to detect LEAD in patients on dialysis was 29.9%, which suggests that the test is not sensitive enough [30]. Therefore, the optimal ABI cutoff value for patients on dialysis is less than 1.06, unlike that in the general population [22]. Since the narrowing of arteries at the peripheral level is more pronounced in patients with renal failure and diabetes mellitus, peripheral microcirculation tests, such as the TBI and skin perfusion pressure, are more useful than the ABI for the early detection of LEAD. To assess peripheral circulatory disturbance, the TBI has a sensitivity of 45.2% and specificity of 100% when using a cutoff TBI value of < 0.6. However, when a cutoff of ABI of <1.06 is used, the sensitivity is 80.0% and specificity 98.0%, indicating that the ABI is a more useful predictor of LEAD [22]. Meanwhile, the TBI is more useful when compared to the ABI in cases of significant arterial calcification in the lower extremities, as it correlates with vascular calcification and LEAD progression [22]. However, cognitive dysfunction in patients with chronic renal failure is thought to be influenced by small vessel dementia, particularly in patients with lacunar infarction associated with small vessel atherosclerosis [36]. The TBI is important in LEAD diagnosis due to arterial calcification; however, it may not be useful in relation to small vessel dementia with atherosclerosis. The subgroup analysis also illustrated that in patients on dialysis without diabetes, the ABI and TBI are significant factors in the MoCA-J and may be useful in detecting MCI. However, in patients on dialysis with underlying diabetes, the results suggest that the ABI and TBI are not significant factors associated with MCI. Although LEAD was shown to be associated with MCI in dialysis patients because of the complex association of many cardiovascular risk factors with factors of mild cognitive impairment in dialysis patients, LEAD was not the only risk factor for MCI. Therefore, patients diagnosed with LEAD based on their ABI values may also have MCI, as assessed by the MoCA-J scores. Not only is regular ABI testing effective in the early diagnosis of LEAD, but it can also simultaneously highlight the risk for MCI. In short, we believe that the ability to assess the risk of MCI through a noninvasive, simple, rapid, and objective method of testing, such as the ABI, is useful for efficient screening in the early stages of MCI.

### Limitations

Since this was a cross-sectional study, it was difficult to infer a causal relationship between each factor, or therapeutic intervention and MCI. In addition, there were insufficient demographic and environmental data (e.g., family structure, diet, and work) related to the patients’ MCI, as well as data on the baseline cognitive function. Furthermore, this study was a single-center study with a small sample size and limited patient selection. The study limitations could be addressed by conducting longitudinal, multicenter cohort studies. The strength of this study is that LEAD detected in the ABI of patients on dialysis has been suggested to be associated with MCI. The validation of this clinically simple index for the detection of MCI in patients on dialysis is a practical contribution to this field. 

## 5. Conclusions

This study showed that the ABI, a marker of LEAD, may be a factor associated with MCI when compared to the TBI. The results indicate the possibility that patients with LEAD with ABI values of less than 1.06 may be at risk for MCI and should therefore be managed accordingly. Future studies are warranted to examine the longitudinal association between LEAD progression and cognitive decline.

## Figures and Tables

**Figure 1 jcm-12-02145-f001:**
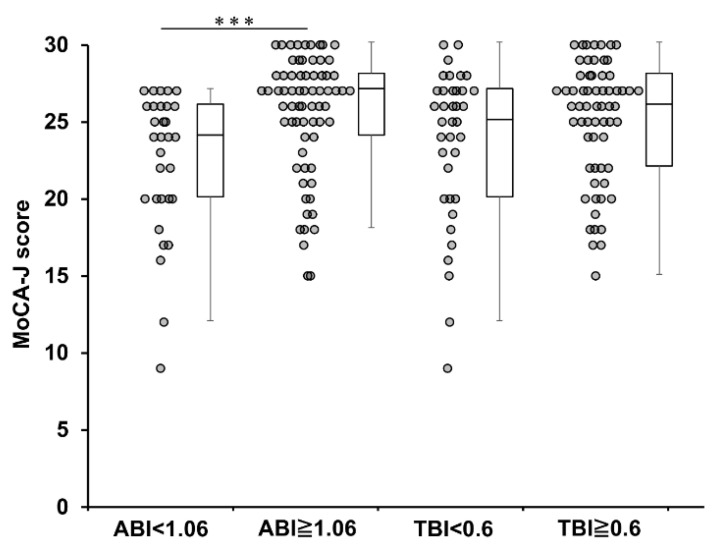
Comparison of MoCA-J score between the two groups divided by ABI and TBI cutoff values. MoCA-J: the Japanese version of the Montreal Cognitive Assessment, ABI: ankle brachial index, TBI: toe brachial pressure index, *** *p* < 0.001.

**Figure 2 jcm-12-02145-f002:**
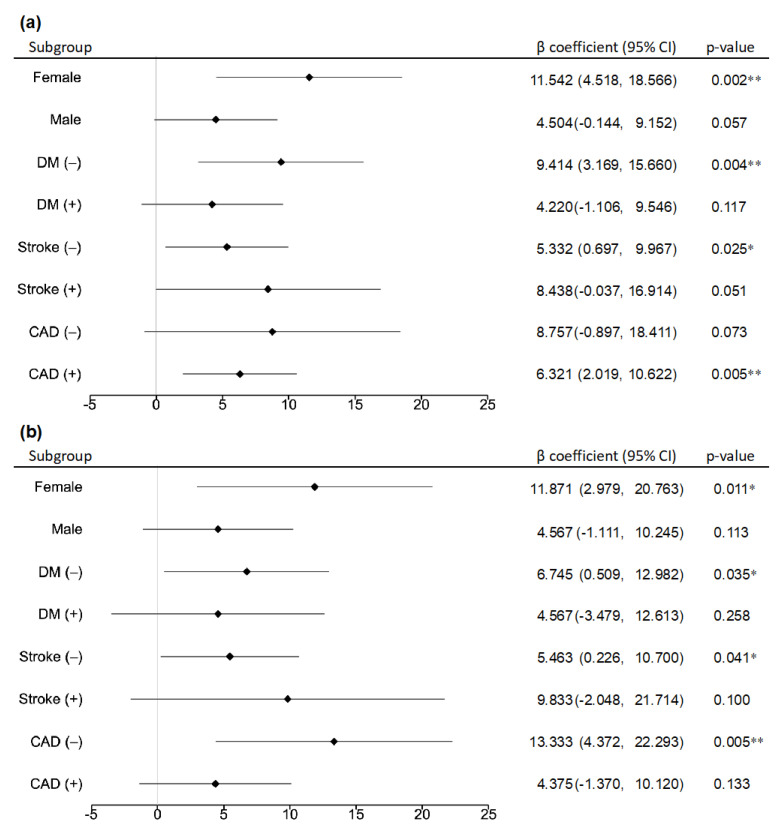
Subgroup analysis of the association between (**a**) ABI/(**b**) TBI and MoCA-J score. The black diamond-shaped symbols represent *β* coefficients (partial regression coefficients), and the black bars indicate 95% confidence intervals. ABI: ankle brachial pressure index, TBI: toe brachial pressure index, MoCA-J: the Japanese version of the Montreal Cognitive Assessment, DM: diabetes mellitus, Stroke: cerebral hemorrhage and cerebral infarction, CAD: coronary arterial disease, * *p* < 0.05, ** *p* < 0.01.

**Table 1 jcm-12-02145-t001:** Characteristics of the participants.

	All Participants *(n* = 100)		MCI (*n* = 46)		No-MCI (*n* = 54)		*p*-Value
Sex [*n*]							
Female	33	(33.0%)	12	(26.1%)	21	(38.9%)	0.175 ^b^
Male	67	(67.0%)	34	(73.9%)	33	(61.1%)	
Age [year]	67.9 ± 11.2	(39–91)	72.9 ± 11.1	(39–91)	63.5 ± 9.3	(43–85)	<0.001 ^c^
BMI [kg/m^2^]	22.9 ± 4.2	(13.4–38.2)	22.3 ± 4.8	(13.4–38.2)	23.4 ± 3.7	(16.1–31.4)	0.104 ^d^
Primary disease [*n*]							
DM	44	(44.0%)	26	(56.5%)	18	(33.3%)	0.064 ^e^
CGN	24	(24.0%)	12	(26.1%)	12	(22.3%)	
Nephrosclerosis	9	(9.0%)	3	(6.5%)	6	(11.0%)	
ADPKD	7	(7.0%)	2	(4.3%)	5	(9.3%)	
IgAGN	5	(5.0%)	0	(0.0%)	5	(9.3%)	
Others	11	(11.0%)	3	(6.5%)	8	(14.8%)	
Complications [*n*]							
CAD	53	(53.0%)	24	(52.2%)	29	(53.7%)	0.476 ^e^
CH	2	(2.0%)	1	(2.2%)	1	(1.9%)	
CI	1	(1.0%)	1	(2.2%)	0	(0.0%)	
Overlapping ^a^	21	(21.0%)	12	(26.1%)	9	(16.7%)	
None	23	(23.0%)	8	(17.4%)	15	(27.8%)	
Duration of maintenance hemodialysis [year]	7.3 ± 6.8	(0–36)	7.6 ± 7.5	(0–36)	7.1 ± 6.3	(0–23)	0.925 ^d^
MoCA-J [score, range: 0–30]	24.6 ± 4.4	(9–30)	20.9 ± 3.8	(9–25)	27.7 ± 1.4	(26–30)	<0.001 ^d^
Hand grip strength [kg]	22.8 ± 7.2	(8.9–39.7)	20.4 ± 7.0	(8.9–37.1)	24.8 ± 6.8	(10.3–39.7)	0.002 ^c^
ABI	1.09 ± 0.21	(0.28–1.36)	1.03 ± 0.26	(0.28–1.35)	1.15 ± 0.14	(0.62–1.36)	0.023 ^d^
Less than 1.06	31	(31.0%)	21	(45.7%)	10	(18.5%)	0.003 ^b^
More than 1.06	69	(69.0%)	25	(54.3%)	44	(81.5%)	
TBI	0.64 ± 0.18	(0.25–0.97)	0.60 ± 0.17	(0.27–0.89)	0.67 ± 0.17	(0.25–0.97)	0.048 ^d^
Less than 0.6	37	(37.0%)	19	(41.3%)	18	(33.3%)	0.411 ^b^
More than 0.6	63	(63.0%)	27	(58.7%)	36	(66.7%)	
PP [mmHg]	66.2 ± 15.2	(29–112)	70.5 ± 14.0	(29–100)	62.5 ± 15.4	(31–112)	0.007 ^c^
Pulse rate [bpm]	72.2 ± 12.9	(45–104)	69.3 ± 12.6	(45–102)	74.8 ± 12.8	(49–104)	0.033 ^c^
ALB [g/dL]	3.5 ± 0.4	(2.1–4.3)	3.3 ± 0.3	(2.1–4.1)	3.6 ± 0.3	(2.7–4.3)	<0.001 ^d^

Values are *n* (percentage), mean ± standard deviation (range), and *p*-value. MCI: mild cognitive impairment, BMI: body mass index, DM: diabetes mellitus, CGN: chronic glomerulonephritis, ADPKD: autosomal dominant polycystic kidney disease, IgAGN: immunoglobulin A glomerulonephritis, CAD: coronary arterial disease, CH: cerebral hemorrhage, CI: cerebral infarction, MoCA-J: the Japanese version of the Montreal Cognitive Assessment, ABI: ankle brachial pressure index, TBI: toe brachial pressure index, PP: pulse pressure, ALB: serum albumin, ^a^: complicated by two or more coronary arterial diseases, cerebral hemorrhage, or cerebral infarction, ^b^: χ^2^ test, ^c^: Student’s t-test, ^d^: Wilcoxon rank-sum test, ^e^: Fisher’s exact test.

**Table 2 jcm-12-02145-t002:** Comparison of MoCA-J scores in two groups divided by (a) ABI and (b) TBI cutoff values.

(a)			
MoCA-J total and subcategories	ABI < 1.06 (*n* = 31)	ABI ≥ 1.06 (*n* = 69)	*p*-value
Total score	22.3 ± 4.6	25.5 ± 3.9	<0.001 ***
Visuoconstructional skills	4.2 ± 1.1	4.6 ± 0.8	0.045 *
Naming task	2.9 ± 0.4	2.9 ± 0.3	0.669
Attention	5.0 ± 1.5	5.5 ± 1.0	0.033 *
Language	1.4 ± 0.8	1.8 ± 0.9	0.023 *
Conceptual thinking	1.4 ± 0.8	1.8 ± 0.5	0.023 *
Memory	1.6 ± 1.5	2.7 ± 1.6	0.001 **
Orientation	5.4 ± 1.2	5.7 ± 0.5	0.188
(b)			
MoCA-J total and subcategories	TBI < 0.6 (*n* = 37)	TBI ≥ 0.6 (*n* = 63)	*p*-value
Total score	23.6 ± 5.0	25.1 ± 3.9	0.173
Visuoconstructional skills	4.3 ± 1.0	4.6 ± 0.8	0.152
Naming task	2.9 ± 0.4	2.9 ± 0.3	0.430
Attention	5.2 ± 1.3	5.4 ± 1.1	0.326
Language	1.5 ± 0.9	1.7 ± 0.9	0.173
Conceptual thinking	1.7 ± 0.6	1.6 ± 0.6	0.820
Memory	2.1 ± 1.6	2.5 ± 1.6	0.377
Orientation	5.5 ± 1.1	5.7 ± 0.6	0.333

MoCA-J: the Japanese version of the Montreal Cognitive Assessment, ABI: ankle brachial pressure index, TBI: toe brachial pressure index, * *p* < 0.05, ** *p* < 0.01, *** *p* < 0.001.

**Table 3 jcm-12-02145-t003:** Estimation of MoCA-J score and ABI/TBI values association in uni- and multivariate linear regression analysis.

Explanatory Variable	Univariate		Multivariate (Stepwise Methods) ^†^	
	*β* Coefficient (95% CI)	*p*-Value	*β* Coefficient (95% CI)	*p*-Value
ABI	6.940 (3.083, 10.797)	< 0.001	4.742 (1.411, 8.073)	0.006 **
Age	―	―	−0.125 (−0.207, −0.044)	0.003 **
Sex	―	―	−1.723 (−3.864, 0.418)	0.113
BMI	―	―	−0.162 (−0.353, 0.029)	0.095
Hand grip strength	―	―	0.190 (0.028, 0.351)	0.022 *
PP	―	―	−0.035 (−0.082, 0.013)	0.154
ALB	―	―	1.795 (−0.344, 3.934)	0.099
Explanatory variable	Univariate		Multivariate (stepwise methods) ^†^	
	*β* coefficient (95% CI)	*p*-value	*β* coefficient (95% CI)	*p*-value
TBI	7.127 (2.378, 11.876)	0.004	4.656 (0.569, 8.743)	0.026 *
Age	―	―	−0.125 (−0.208, −0.041)	0.004 **
Sex	―	―	−1.989 (−4.174, 0.196)	0.074
BMI	―	―	−0.147 (−0.341, −0.046)	0.040 *
Hand grip Strength	―	―	0.220 (0.057, 0.382)	0.009 **
ALB	―	―	1.837 (−0.365, 4.039)	0.101
Explanatory variable	Multivariate (forced entry methods) ^††^			
	*β* coefficient (95% CI)	standardized *β* coefficient (95% CI)		*p*-value
ABI	5.310 (0.566, 10.054)	0.260 (0.028, 0.492)		0.029 *
TBI	3.376 (−2.362, 9.113)	0.136 (−0.095, 0.368)		0.246

^†^ Explanatory variables entered into multivariate regression using stepwise methods (ABI/TBI, sex, age, body mass index, duration of maintenance hemodialysis, hand grip strength, pulse pressure, pulse rate, serum albumin), ^††^ Explanatory variables entered into multivariate regression using forced entry methods (ABI, TBI), MoCA-J: the Japanese version of the Montreal Cognitive Assessment, ABI: ankle brachial pressure index, TBI: toe brachial pressure index, BMI: body mass index, PP: pulse pressure, ALB: serum albumin, * *p* < 0.05, ** *p* < 0.01.

## Data Availability

The data presented in this study is available from the corresponding author on request.

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
