# Peer review of "Relationship between Lower Extremity Peripheral Arterial Disease and Mild Cognitive Impairment in Hemodialysis Patients"

_jcm, 2023, doi:10.3390/jcm12062145_

Round 1
Reviewer 1 Report
In this manuscript, the authors evaluated the relationship between LEAD, assessed by ABI and TBI, and MCI, assessed by MOCA-J, in HD patients. The results showed that 46% patients were diagnosed with MCI, and that ABI values were significantly and positively correlated with MOCA-J scores and were also superior to TBI values. I think that this study is interesting, although there are several points should be revised.
1. Analytic methods: Although multiple regression models by step wise methods were used and conducted with MOCA-J scores as the objective variable in this manuscript, explanatory variables were included ABI/TBI, gender, age, BMI, myocardial infarction, cerebral hemorrhage, cerebral infarction, history of diabetes, HD vintage, hand grip strength, pulse pressure, pulse rate, and serum albumin levels. I think that the number of explanatory variables should be less than total case number (N=100) / 10 in statistical analyses (less than 10 including ABI/TBI).
2. The authors should discuss why ABI was superior to TBI in the association with MCI, more precisely.
Author Response
Please see the attachiment.

Reviewer 2 Report
In this retrospective, cross-sectional study Nishimura et al showed that ankle brachial index (ABI) can be associated with mild cognitive impairment (MCI) when compared to toe brachial Index (TBI). Although the study wants to investigate a potentially interesting topic of the association between MCI and ABI, the retrospective nature and some imperfections in the methodological structure have not allowed to acquire a substantial novelty nor significant scientific advancement. below are some of the major and minor issues that can be found in the study:
1. Table 1 lists the causes of primary disease and diabetes mellitus appears to be the predominant cause. Making a distinction only based on the causes of nephropathy it is not clear whether it also means the general prevalence of diabetes mellitus in the sample examined or whether some of the patients with other known causes of nephropathy could present diabetes as a comorbidity. Please provide this information.
2. The choice of including a sample with a heterogeneous etiology of nephropathy is questionable. Probably a separate analysis in the subgroup of diabetic patients who are at higher risk for LEAD and MCI could provide more appeal and strength to the study
3.the cut offs chosen for the ABI take into consideration only values lower or higher than 1.06. Given the population of nephropathic patients and the high percentage of diabetics, it is probable that there is a high prevalence of calcified lesions of the lower limbs which may be responsible for ABI values >1.4. It is therefore not clear whether this subgroup of arteriopathic patients was excluded as it was not mentioned among the exclusion criteria or whether patients with ABI > 1.4 were included among patients with ABI > 1.06 and therefore considered to have mild atherosclerotic disease.
4. it would have been appropriate to delve a little more into the prognostic significance of LEAD and the difference also in terms of classification with PAD
5. Some periods are a bit too complex and unclear. It would require a native speaker revision
6.on line 227 the abbreviation PAD appears for the first time but mon has been explained previously. It would also be appropriate to mention the implications of the PAD
7.the final part of the discussion and the conclusions highlighting the potential implications of the study about consider the patient's cognitive function when teaching foot care in order to prevent LEAD appears a bit forced and does not have a particular clinical impact
Round 2
Reviewer 2 Report
The authors addressed sufficiently the raised issues.